# Conformational ensemble of the human TRPV3 ion channel

Lejla Zubcevic [1], Mark A. Herzik Jr. [2], Mengyu Wu[2,3], William F. Borschel[1], Marscha Hirschi [1,4], Albert S. Song [2,3], Gabriel C. Lander [2] & Seok-Yong Lee [1]

Transient receptor potential vanilloid channel 3 (TRPV3), a member of the thermosensitive TRP (thermoTRPV) channels, is activated by warm temperatures and serves as a key regulator of normal skin physiology through the release of pro-inflammatory messengers. Mutations in *trpv3* have been identified as the cause of the congenital skin disorder, Olmsted syndrome. Unlike other members of the thermoTRPV channel family, TRPV3 sensitizes upon repeated stimulation, yet a lack of structural information about the channel precludes a molecular-level understanding of TRPV3 sensitization and gating. Here, we present the cryo-electron microscopy structures of apo and sensitized human TRPV3, as well as several structures of TRPV3 in the presence of the common thermoTRPV agonist 2-aminoethoxydiphenyl borate (2-APB). Our results show α-to-π-helix transitions in the S6 during sensitization, and suggest a critical role for the S4-S5 linker π-helix during ligand-dependent gating.

[1] Department of Biochemistry, Duke University Medical Center, Durham 27710 NC, USA. [2] Department of Integrative Structural and Computational Biology, The Scripps Research Institute, La Jolla 92037 CA, USA. [3] Skaggs Graduate School of Chemical and Biological Sciences, The Scripps Research Institute, La Jolla 92037 CA, USA. [4]Present address: Department of Integrative Structural and Computational Biology, The Scripps Research Institute, La Jolla 92037 CA, USA. These authors contributed equally: Lejla Zubcevic, Mark A. Herzik Jr., Mengyu Wu. Correspondence and requests for materials should be addressed to G.C.L. (email: glander@scripps.edu) or to S.-Y.L. (email: seok-yong.lee@duke.edu)

Transient receptor potential (TRP) channels are a super-family of cation-selective ion channels that are involved in numerous physiological processes[1]. The vanilloid TRP (TRPV) subfamily consists of six members (TRPV1−TRPV6), four of which (TRPV1−TRPV4) possess an intrinsic capability to respond to temperature and are therefore referred to as ther-mosensitive TRPV (thermoTRPV) channels. In addition to heat, thermoTRPV channels are also modulated by synthetic and natural ligands as well as lipids[2−8]. Although the thermoTRPV channels share >50% homology, each channel possesses a distinct functional and pharmacological profile that enables integration of a multitude of different stimuli into a finely tuned response[2,7,9,10].

In contrast to other thermoTRP channels, TRPV3 sensitizes, rather than desensitizes, upon repeated stimulation with either heat or agonists[11−13]. Notably, TRPV3 sensitization is inherent to the channel and independent of the modality of the stimulus. The channel undergoes hysteresis, which irreversibly lowers the energetic barrier for activation, resulting in a faster rate of acti-vation and higher open probability[13]. TRPV3 is most abundantly expressed in epidermal and hair follicle keratinocytes where it plays a key role in the maintenance of normal skin physiology[11,14]. Activation of TRPV3 in keratinocytes has been shown to lead to release of proinflammatory messengers, and several "gain-of-function" mutations in *trpv3* have been found to cause the human congenital skin disorder Olmsted syndrome, characterized by bilateral mutilating palmoplantar keratoderma, periorificial keratotic plaques, and hair loss with follicular papules[15−22]. Therefore, TRPV3 plays essential roles in epidermal proliferation, differentiation, survival, hair growth, and the development of itch sensation[23]. Several small-molecule inhibi-tors, such as FTP-THQ and GRC 15300, of TRPV3 have demonstrated analgesic properties in inflammatory and neuro-pathic pain models[24,25].

High-resolution structures of thermoTRPV channels captured in various states along the gating cycle have been critical to understanding the molecular underpinnings that govern both general and individual functional modalities of thermoTRPV channels[26−30]. However, as the structure of TRPV3 has yet to be determined, a large gap currently exists in our mechanistic understanding of thermoTRPV channel sensitization in activation and regulation. To fill this knowledge gap, we determined the cryo-EM structures of human TRPV3 in apo and sensitized states at ~3.4 Å and ~3.2 Å resolution, respectively. We additionally determined three conformationally distinct states of TRPV3 in the presence of the agonist 2-aminoethoxydiphenyl borate (2-APB), at resolutions ranging from ~3.5–4 Å, where we observe a departure from the canonical fourfold symmetry previously reported for most thermoTRPV structures. Together, these structures illustrate the importance of π-helical segments in the transmembrane domain of TRPV3 in dictating the conforma-tional landscape during sensitization and gating.

## Results

**Functional characterization of human TRPV3.** Extensive screening of various TRPV3 homologs and mutants identified human TRPV3 containing the point mutation T96A as optimal for structural studies (see Methods). Characterization of both T96A and wild-type channels confirmed that the T96A mutation did not significantly alter the ligand-dependent gating profile of the TRPV3 channel (Fig. 1c–g). Whole-cell electrophysiological recordings show that TRPV3 T96A channels are functionally active and sensitize to repeated applications of the agonist 2-APB (30 μM) with a sigmoidal time course of sensitization, similar to wild-type channels (Fig. 1c, d). There were no significant differ-ences between sensitization parameters for both the time course

and extent of sensitization for wild-type and T96A channels, indicating the T96A mutation does not alter agonist-induced use-dependence of gating and subsequent sensitization of TRPV3 (Fig. 1e–g).

**Architecture of human TRPV3.** We determined the structure of full-length, human TRPV3$_{T96A}$, hereafter simply referred to as TRPV3, in the closed apo state to ~3.4 Å using cryo-electron microscopy (cryo-EM) (see Methods) (Figs. 1a, 2 and Table 1). The TRPV3 channel, like other thermoTRPV channels (TRPV1, TRPV2, and TRPV4[27−30]), is a fourfold symmetric homo-tetramer. Each monomer contains an amino- (N-) terminal cytosolic ankyrin repeat domain (ARD) and a domain-swapped transmembrane domain (TMD) comprising six transmembrane helices (S1−S6). The TMD and ARD are connected via the coupling domain (CD), which consists of the linker domain, pre-S1, and the carboxy- (C-) terminal domain (CTD). The TMD consists of a voltage sensor-like domain (VSLD) containing helices S1–S4, and a pore domain comprising helices S5, S6 and the pore helix. The VSLD and the pore domain are connected via an S4−S5 linker, which mediates the domain-swap configuration. In addition, the channel contains a TRP domain (Fig. 1b). The stabilizing T96A mutation, located N-terminal to the ARD (residues 1–117), was not resolved in our 3D reconstructions (Supplementary Fig. 1).

Comparison of our closed, apo TRPV3 structure to the previously determined apo structures of the TRPV1[27] (PDB 3J5P), TRPV2[28] (PDB 5AN8) and TRPV4[30] (PDB 6BBJ) channels reveals several notable features. First, the high quality of the TRPV3 EM density map has allowed us to build the most complete thermoTRPV CTD to date (residues 707–754), a region that was not well resolved in the cryo-EM structures of TRPV1, TRPV2, or TRPV4[26−30]. The most N-terminal region of the CTD, which extends from the TRP domain into the cytosol, forms a short α-helix similar to that observed in TRPV4[30], but in contrast to the loops observed in TRPV1 and TRPV2[26,27,29] (Fig. 1b). The CTD extends into the cytosol, forming a short two-stranded β-sheet before adding an additional strand to the β-sheet of the CD. In contrast to previously characterized TRPV channel structures, the C-terminal end of the CTD encircles the CD and loops across the peripheral side of the CD to interact with the ARD of the neighboring protomer (Fig. 3a). This extended CTD establishes previously unobserved intra- and inter-subunit interactions that may serve a functional role (see "The CTD mediates inter-protomer contacts" section for further discussion). Second, the selectivity filter (SF, residues 636–640) of TRPV3 is lined with backbone carbonyls, whereas the SFs of TRPV1 and TRPV2 are occluded by hydrophobic amino acid side-chains, such as methionine. Similarly to the SF in TRPV4, the TRPV3 SF is sufficiently wide (~7 Å) to permeate a semi-hydrated calcium ion (Fig. 3b). Third, we previously proposed that the presence of π-helices in the strategically important S4−S5 linker and S6 in TRPV channels likely acts as transient flexible hinge points that facilitate channel gating[28]. Indeed, apo TRPV3 possesses a π-helix in the S4−S5 linker, whereas S6 is fully α-helical (Fig. 3c).

**The sensitized state of the TRPV3 channel.** In order to elucidate the molecular mechanism underlying TRPV3 sensitization, we determined the ~3.2 Å cryo-EM structure of TRPV3 that had been treated with the agonist 2-APB according to a sensitization protocol developed in-house and guided by electrophysiological results (see Methods, Fig. 1c, d, Supplementary Fig. 2). Com-parison of this putative sensitized structure to the apo state did not reveal any substantive structural changes, with a 1.2 Å Cα root mean square deviation (RMSD). However, closer inspection

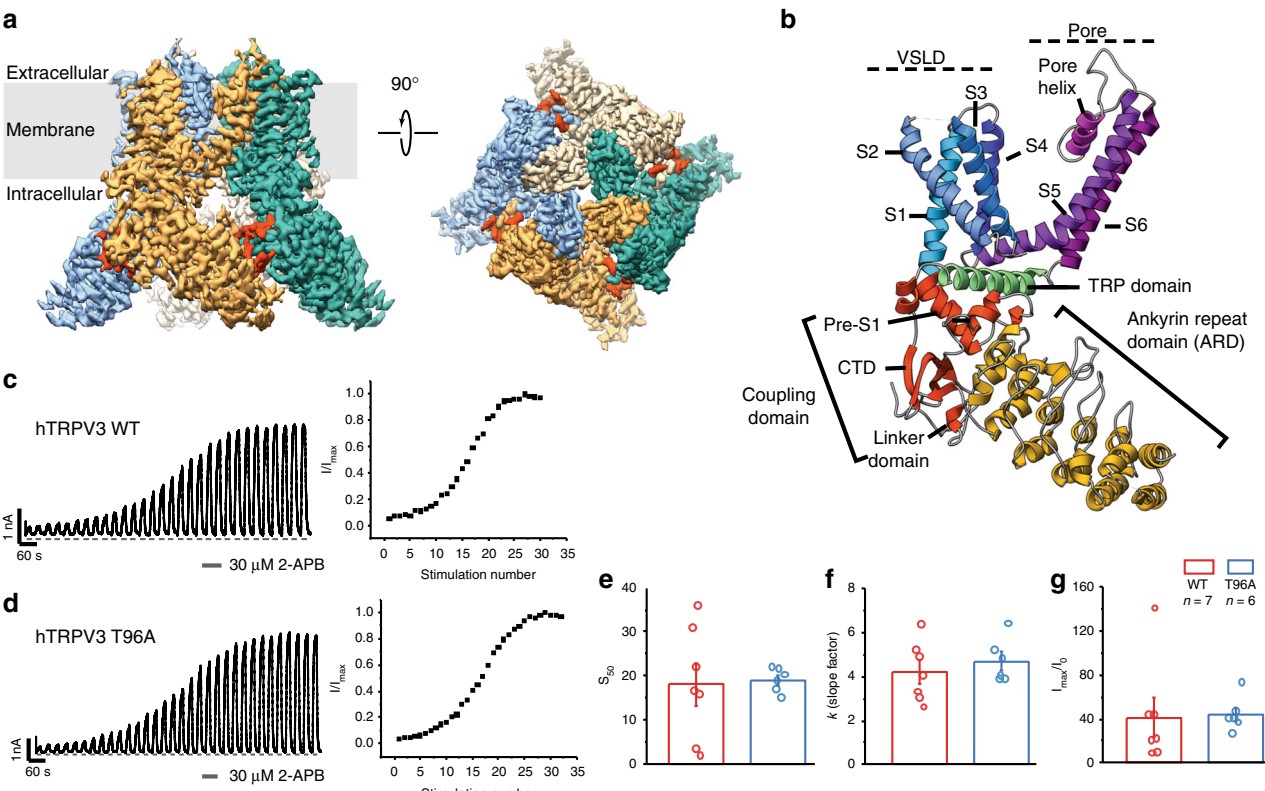

**Fig. 1** Structural and functional characterization of the human TRPV3. **a** 3D cryo-EM reconstruction of the apo human TRPV3 tetramer, colored by protomer. The C-terminal domain is shown in red. **b** Overview of the structural elements within a single protomer of the TRPV3 channel: Ankyrin repeat domain (ARD) is shown in yellow; Coupling domain (CD), consisting of linker domain, pre-S1 and the C-terminal domain, is shown in red; Voltage sensing-like domain (VSLD), consisting of helices S1—S4, is shown in blue, the pore domain, consisting of helices S5—S6 and the pore helix, is colored in purple; the TRP domain is shown in green. **c**, **d** Functional characterization of the TRPV3 wild-type (**c**) and T96A mutant (**d**) channels. Representative whole-cell current traces recorded at +60 mV from TRPV3 wild-type and T96A channels evoked by repeated application of 30 μM 2-APB for 15 s followed by complete washout (left) and corresponding time course of sensitization from the depicted recording (right) (TRPV3WT: $n = 7$ biologically independent experiments; TRPV3 T96A: $n = 6$ biologically independent experiments). **e**, **f** The half-maximal stimulation number ($S_{50}$) (**e**) and rate of change ($k$) (**f**) of sensitization were calculated as the mean values obtained from fits of time course of sensitization plots from individual recordings (TRPV3WT: $n = 7$ biologically independent experiments; TRPV3 T96A: $n = 6$ biologically independent experiments) with standard two-state Boltzmann equation (see Methods). The extent of sensitization (**g**) was characterized by the relative increase in the current response to 2-APB during the first ($I_0$) and maximum current ($I_{max}$) stimulation ($I_{max}/I_0$) and calculated as the mean from each biologically independent experiment. No significant differences (NS) in the $S_{50}$ ($P = 0.86$), $k$ ($P = 0.46$), and $I_{max}/I_0$ ($P = 0.90$) was determined between TRPV3WT and TRPV3 T96A (two-tailed Student's $t$ test, $P > 0.05$). Confidence intervals (95%): −14.2 (low)/12.2 (high) for $S_{50}$, −2.54 (low)/1.52(high) for $k$, −55.1 (low)/49.8 (high) for $I_{max}/I_0$. Bar graphs and error bars denote mean ± s.e.m. The source data underlying Fig. 1e–g are provided as a Source Data file

of the pore revealed that our 2-APB sensitization protocol resulted in the formation of a π-helix in S6 and a consequent change in register of the pore-forming residues (Fig. 4a). Specifically, while the α-helical S6 in apo TRPV3 imposes a narrow constriction at residue M677, the putative sensitized TRPV3 contains a π-helix that begins at residue F666 and changes the register of the S6 helix, resulting in a constriction at I674 (Fig. 4a, b). In addition, the C-terminal half of the S6 helix in this structure is bent at a ~9° angle from the inner pore when compared to the S6 in the apo structure (Fig. 4c). This bend is caused by the disruption of interactions between the S6 and the S4—S5 linker of the neighboring protomer, in the vicinity of the vanilloid binding site in TRPV1[10,26,31,32]. In the apo structure, where the S6 helix is entirely α-helical, two pairs of interactions are present between S6 and the S4—S5 linker: one between residue N671 in S6 and the backbone carbonyl of Y575 located in the S4—S5 linker, and the other between the hydroxyl group of Y575 and the backbone amide of M672 within S6. Together, these interactions between S6 and the S4—S5 linker stabilize a straight, α-helical conformation of S6 (Fig. 5a). The register shift in S6 of the putative sensitized

structure disrupts these interactions, releasing S6 and allowing it to bend away from the S4—S5 linker. In this structure, residue N671 faces the pore and appears to stabilize the π-helix by interacting with the backbone carbonyl of V667 (Fig. 5b).

Notably, N671 is the only polar residue in the S6 helix of TRPV3 (Fig. 5c) and we observe here that it adopts distinct positions depending on the functional state of the channel: either facing the S4–S5 linker in the apo state, or facing the pore in the sensitized state. This suggests that, in addition to introducing a flex point into S6, the α-to-π transition also decreases the hydrophobicity of the pore and provides a potential ion coordination site. The high sequence conservation of N671 in TRP channels (Fig. 5c) suggests that its functional and structural roles might also be conserved. Indeed, molecular dynamics simulations and free energy calculations of the TRPV1 pore have suggested a role for N676, which corresponds to N671 in TRPV3, in hydration of the pore[33,34]. These observations support our prior proposal that state-dependent α-to-π transitions in S6 occur in thermoTRPV channels[28], and reflect the α-to-π transition described for TRPV6 channel opening[35]. However, while the

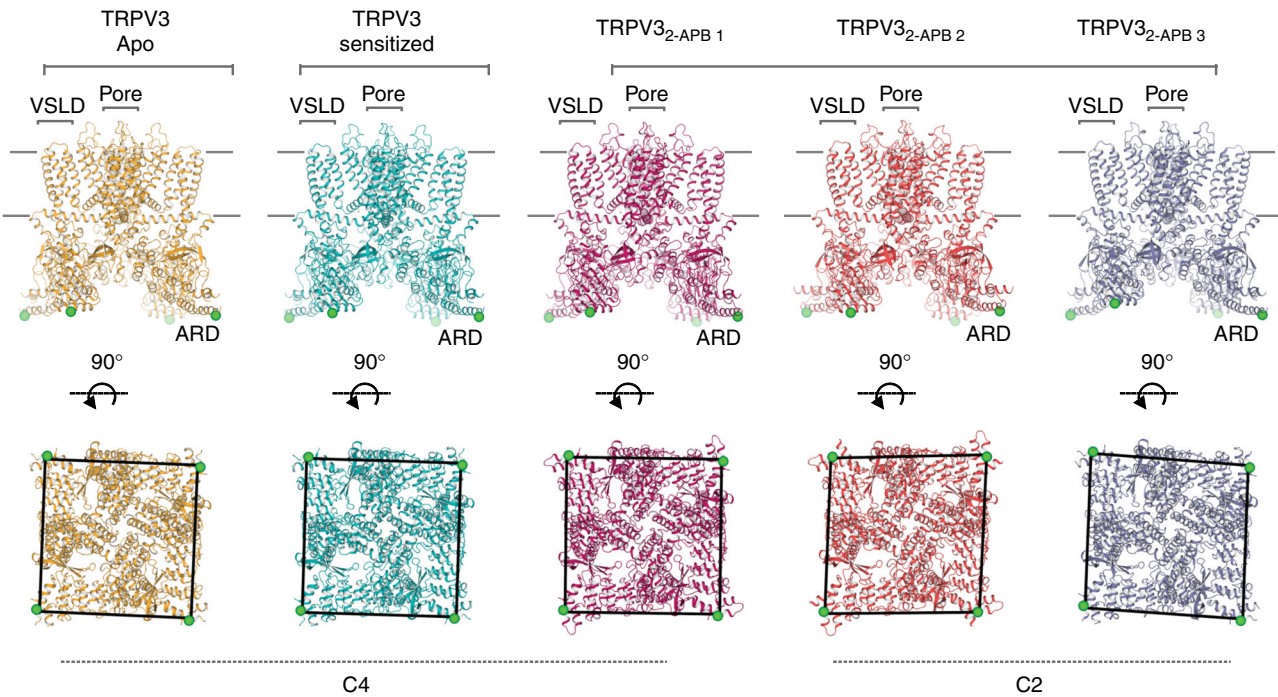

**Fig. 2** The conformational ensemble of human TRPV3. Overview of TRPV3 structures: apo (gold), sensitized (cyan), TRPV3$_{2-APB\ 1}$ (magenta), TRPV3$_{2-APB\ 1}$ (red), TRPV3$_{2-APB\ 3}$ (purple) (from left to right). Top panel shows the view from the membrane plane and the bottom panel represents the bottom-up view. Green sphere represents residue 145 and illustrates the positions of subunits relative to one another. Voltage sensor-like domain (VSLD), pore and ankyrin repeat domain (ARD) are labeled

transition in TRPV6 was induced by a mutation, the transition from a low-energy α- to a high-energy π-helix in TRPV3 is ligand-induced.

**TRPV3 departs from fourfold symmetry when 2-APB is present.** To elucidate the conformational changes governing transition of TRPV3 from the closed to the open state, we determined structures of the channel in the presence of the agonist 2-APB (TRPV3$_{2-APB}$, see Methods). A single cryo-EM dataset yielded three distinct conformations, TRPV3$_{2-APB\ 1}$, TRPV3$_{2-APB\ 2}$, and TRPV3$_{2-APB\ 3}$ (Fig. 2, Supplementary Fig. 3, Supplementary Fig. 4), all of which resolved to ~4 Å resolution or better. Strikingly, two out of the three TRPV3$_{2-APB}$ structures exhibit twofold (C2) symmetry (TRPV3$_{2-APB\ 2}$, and TRPV3$_{2-APB\ 3}$, Fig. 2), deviating from the canonical fourfold (C4) symmetry previously associated with thermoTRPV channels[26–30]. Notably, a thorough reanalysis of the apo TRPV3 datasets yielded no evidence of structural species containing symmetries lower than C4 (see Supplementary Fig. 1). The reduced symmetry is conspicuous in the pores of the TRPV3$_{2-APB\ 2}$ and TRPV3$_{2-APB\ 3}$ structures, where movement in the C-terminal region of S6 in diagonally opposing protomers results in asymmetric gate arrangements within the channels (Fig. 6a, b). In TRPV3$_{2-APB\ 2}$ (Fig. 6a) the distance between the M677 residues in subunits A and C is ~5.7 Å, similar to the diameter of the closed gate of apo TRPV3 (~5.2 Å). However, in subunits B and D this distance is increased to ~9.1 Å which is similar to the ~9.3 Å diameter measured between gate residues in the structure of TRPV1 bound to double-knot toxin (DkTx) and resiniferatoxin (RTx)[26]. However, TRPV3$_{2-APB\ 2}$ is likely to be nonconductive, as the narrow opening of subunits A and C constricts the pore (Fig. 6b). It is thus possible that this arrangement of subunits represents an intermediate state. The twofold symmetry is present not only in the pore of TRPV3$_{2-APB}$, but also across the entire channel (Fig. 2).

To better understand how these states deviate from the canonical C4-symmetric organization, we overlaid the sensitized TRPV3 structure with that of TRPV3$_{2-APB\ 2}$. Due to the domain-swapped configuration of the TMDs, each subunit is composed of the VSLD of one protomer and the pore domain of the neighboring protomer. In this arrangement, the S4−S5 linker connects each subunit to its neighbor (Fig. 7a). The overlay of the sensitized TRPV3 structure with that of TRPV3$_{2-APB\ 2}$ indicates that while TMDs of subunits A and C appear to be nearly identical between the two conformations, the TMDs of subunits B and D in TRPV3$_{2-APB\ 2}$ undergo a ~3° rotation with respect to that of sensitized TRPV3 (Fig. 7a and Supplementary Fig. 5).

C2 symmetry is also present in the ARDs, where two of the ARDs are rotated with respect to the central axis of the pore and the remaining two assume the same conformation as in the sensitized TRPV3 (Supplementary Fig. 5). Surprisingly however, the deviating ARD is the one coupled to subunit B via the ARD–CTD interface, rather than the ARD directly connected to subunit B via the coupling domain (Supplementary Fig. 5). We therefore reason that this ARD is part of subunit B and that ARDs and TMDs are coupled through the ARD–CTD interface.

A rigid body superposition of the B subunits from the sensitized TRPV3 and TRPV3$_{2-APB\ 2}$ structures results in good alignment (Cα RMSD = 0.9 Å), with only the S4−S5 linker deviating significantly. This suggests that the S4−S5 linker is the origin of the rotation observed in subunit B of TRPV3$_{2-APB\ 2}$ (Fig. 7b). Different conformations of the S4−S5 linker were also observed when subunits A and B of TRPV3$_{2-APB\ 2}$ were superposed (Cα RMSD = 0.9 Å), indicating that this region is the source of C2 symmetry in the TRPV3$_{2-APB\ 2}$ structure (Fig. 7c).

Interestingly, an overlay of S5 helices of subunits A and B revealed that their S4−S5 linkers contain π-helices that originate at two different positions: at residue I583 in subunit A and at residue M578 in subunit B (Fig. 7d). Since the π-helices dictate

**Table 1 Cryo-EM data collection, refinement and validation statistics**

| | Apo hTRPV3 6MHO EMD-9115 | Sensitized hTRPV3 6MHS EMD-9117 | hTRPV3$_{2\text{-APB 1}}$ 6MHV EMD-9119 | hTRPV3$_{2\text{-APB 2}}$ 6MHW EMD-9120 | hTRPV3$_{2\text{-APB 3}}$ 6MHX EMD-9121 |
|---|---|---|---|---|---|
| **Data collection and processing** | | | | | |
| Magnification | ×22,500 | ×45,000 | ×36,000 | | |
| Voltage (kV) | 300 | 200 | 200 | | |
| Electron exposure (e- Å$^{-2}$) | 63 | 67 | 60 | 60 | 60 |
| Defocus range (μm) | −0.5 to −2 | −0.8 to −1.4 | −1.2 to −2 | −1.2 to −2 | −1.2 to −2 |
| Pixel size (Å) | 1.31 | 0.915 | 1.15 | 1.15 | 1.15 |
| Detector | Super-resolution | Counting | Counting | Counting | Counting |
| Total extracted particles (no.) | 361,244 | 559,206 | 897,643 | | |
| Refined particles (no.) | 215,346 | 339,693 | 589,656 | | |
| **Reconstruction** | | | | | |
| Final particles (no.) | 27,620 | 44,554 | 141,832 | 61,292 | 84,307 |
| Symmetry imposed | C4 | C4 | C4 | C2 | C2 |
| **Nominal resolution (Å)** | | | | | |
| FSC 0.143 (unmasked/masked) | 4.21/3.38 | 4.10/3.24 | 3.83/3.45 | 4.98/3.99 | 4.91/3.99 |
| Map sharpening B factor (Å$^2$) | −73 | −51 | −145 | −152 | −144 |
| **Refinement** | | | | | |
| **Model composition** | | | | | |
| Non-hydrogen atoms | 17,032 | 17,784 | 18,336 | 17,892 | 16,956 |
| Protein residues | 2404 | 2428 | 2452 | 2452 | 2436 |
| Ligands | 0 | 0 | 0 | 0 | 0 |
| **Validation** | | | | | |
| MolProbity score | 1.12 | 1.61 | 1.61 | 1.57 | 1.55 |
| Clashscore | 3 | 6 | 7 | 7 | 7 |
| Poor rotamers (%) | 0 | 0 | 0 | 0.1 | 0 |
| **Ramachandran plot** | | | | | |
| Favored (%) | 98 | 96 | 96 | 97 | 97 |
| Allowed (%) | 2 | 4 | 4 | 3 | 3 |
| Disallowed (%) | 0 | 0 | 0 | 0 | 0 |

the angle between subunits, it is likely that the differing positions of S4—S5 π-helices within the tetramer are responsible for the twofold symmetric arrangement of the channel. Because the break from C4 symmetry is only evident in structures obtained in the presence of 2-APB, we posit that the TRPV3 channel can assume reduced symmetry intermediates. Interestingly, a similar departure from C4 symmetry arising from a rearrangement of the S4–S5 linker was observed in the crystal structure of TRPV2 upon binding RTx[36]. Notably, the gain-of-function mutations that cause Olmsted syndrome are located in the S4—S5 linker and its junctions with the S4b, S3, the TRP domain, and the lower part of S6, underscoring the importance of the S4–S5 linker in TRPV3 channel gating (Supplementary Fig. 6).

Although our reconstructions did not enable unambiguous identification of 2-APB, distinct nonpolypeptide densities were observed in both the vanilloid binding pocket as well as in the cavity formed by the VSLD and the TRP domain (Supplementary Fig. 7). The latter was recently identified as the binding site for 2-APB in TRPV6[37], for which it has an inhibitory, rather than activating effect on the channel. However, since both sites are often occupied by putative lipid molecules[26,28,38–40], it is difficult to determine whether the observed densities belong to the bound ligand or a lipid. Furthermore, the density in the VSLD cavity remains consistent in all of our TRPV3 reconstructions.

**The CTD mediates inter-protomer contacts**. Notably, TRPV3 contains an extended CTD that has not been resolved in the previously reported cryo-EM structures of TRPV1, TRPV2, or TRPV4[26–30]. The distal region of the CTD wraps around the β-sheet of the coupling domain at the interface between two

neighboring protomers, where it contacts the ARD fingers 1–3 (Fig. 3a). This region of the ARD has previously been identified as important for channel gating, and has been implicated in binding of calmodulin (CaM) and ATP[41–44]. Furthermore, biochemical and structural studies of isolated ARDs have identified the conserved residue K169 as being critical for ATP binding[42–44]. However, in our structures, we observe that K169 in the ARD of one protomer forms salt bridges with residues E751 and D752 located in the CTD of the adjacent protomer (Supplementary Fig. 8). Therefore, binding of CaM or ATP to the proposed ARD binding site involving K169 would not be possible for the conformations captured in this study, as the CTD and ATP binding sites on the ARDs appear to overlap (Supplementary Fig. 8).

Our structural analyses suggests an important role for the extended CTD in channel gating, specifically in mediating the mechanic coupling of the ARDs and TMs. Interestingly, sequence comparison with other thermoTRPV channels strongly suggests that the extended CTD is a conserved feature in thermoTRPV (Supplementary Fig. 8). In the context of previous studies, it is tempting to speculate that the CTD and CaM/ATP might compete for binding in this region of the ARD and that this competition might be a part of a regulatory mechanism in TRPV3.

## Discussion
We determined the structures of the human TRPV3 channel in multiple conformational states, providing a glimpse of putative gating intermediates. The apo structure of TRPV3, despite being closely related to other thermoTRPV channels, shows a number of unique features. Most notably, in contrast to the selectivity

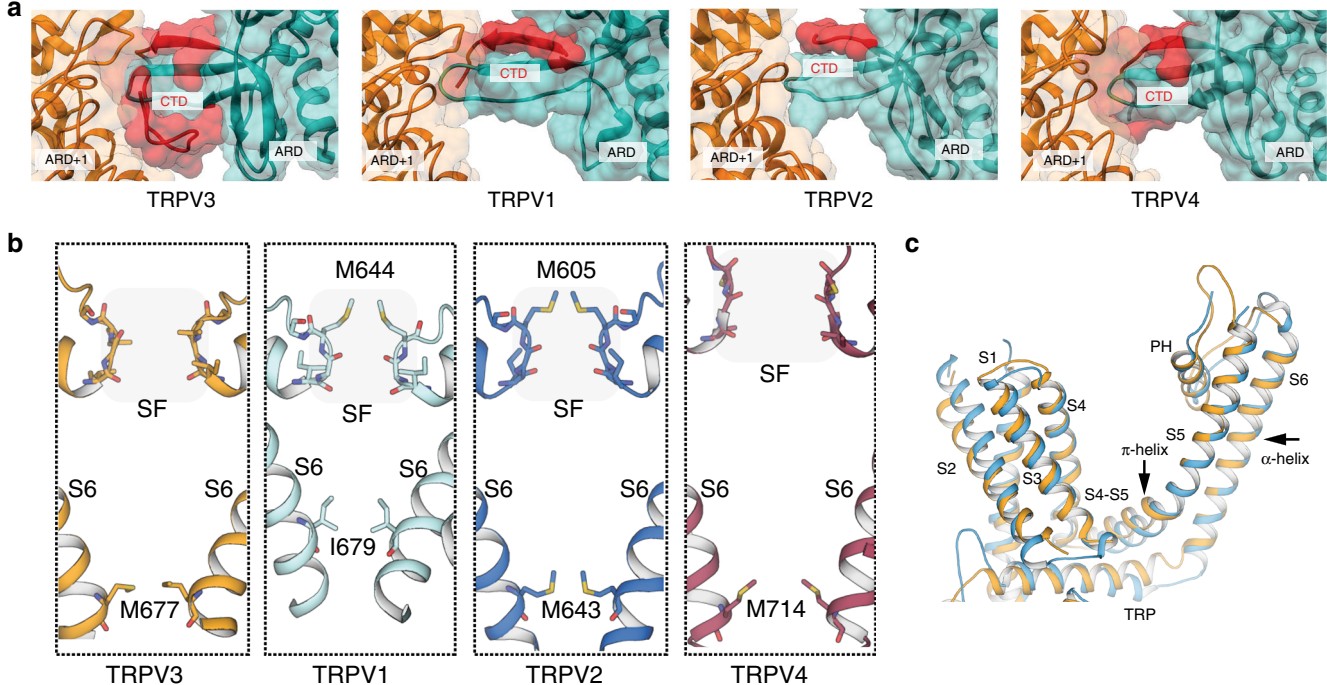

**Fig. 3** Comparison of apo TRPV3 with thermoTRPV channels. **a** Surface representation of the C-terminal domain (CTD) in TRPV3 compared to the CTDs in TRPV1, TRPV2, and TRPV4. The C-terminus of the CTD is resolved in TRPV3 and encircles the coupling domain (CD) β-sheet and forms an interface with the neighboring ARD (ARD$^{+1}$). **b** Comparison of selectivity filters (SF) and helix bundle gates in TRPV3 (gold), TRPV1 (cyan), TRPV2 (blue), and TRPV4 (purple). SF residues are indicated by a shaded box and shown in stick representation. Residues representing the helix bundle gate are shown in stick representation. **c** Overlay of the transmembrane domains (TM) of TRPV2 (marine) and TRPV3 (gold) of a single monomer. The TMs of TRPV2 are all α-helical, while the S4—S5 linker of TRPV3 contains a π-helical turn

filters (SFs) of TRPV1 and TRPV2[27,28], which are occluded by side chains of hydrophobic residues, the SF of TRPV3 is lined with backbone carbonyl groups and is wide enough to permeate partially dehydrated cations. Interestingly, the SF remains mostly unchanged in all the conformational states captured, which suggests that the SF in TRPV3 does not act as a gate, in contrast to TRPV1 and TRPV2.

TRPV3 is unique among TRP channels in that it sensitizes upon repeated application of stimuli. Here we show that exposure of the TRPV3 channel to a chemical sensitization protocol results in a change in the secondary structure of the pore-lining helix S6, which adopts a π-helix. The α-to-π transition in S6 has been proposed to play an important role in TRPV channel gating by introducing a hinge in the S6[28] analogous to glycine and proline hinges in K$^+$ channels[45,46]. However, in contrast to the hinges in K$^+$ channels which are engineered into the amino acid sequence, the π-helix hinge was predicted to form depending on the functional state of the channel[28]. While this manuscript was under review, structures of the mouse TRPV3 (mTRPV3) in apo (closed) and 2-APB bound (open) states were published[47], allowing us to draw comparisons to our data. Most strikingly, the open mTRPV3 channel possesses a π-helix in the S6. We therefore reason that our putative sensitized state, which contains a π-helical segment in S6 and has a closed gate, illustrates the hallmarks of a sensitized channel that is closed but requires less energy for activation than the naïve channel (Supplementary Fig. 9). This leads to the critical insight that the α-to-π transition does not necessitate channel opening, but can result in a closed state that has a lower energy barrier for opening. It is also interesting to point out the apparent correlation between use-dependence and α-to-π transitions in S6. Heat- and ligand-dependent use-dependence has been observed in TRPV2 and

TRPV3, but not in TRPV1[9,13]. While the TRPV2 and TRPV3 channels undergo secondary structure transitions in S6 during opening, the apo/closed TRPV1 channel already possesses a π-helix[27,40]. This is in agreement with our suggestion that α-to-π transition is important for sensitization; however, further study is required to probe the role of secondary structure transitions in this process. Moreover, the putative sensitized state captured here may be one of many, as functional data suggest that TRPV3 may possess multiple closed, sensitized states, with decreasing energy requirements for opening[13]. Furthermore, structural elements other than the S6 might be involved in sensitization of TRPV3[48].

Interestingly, the apo mTRPV3 has a unique arrangement of the pore domain (S5, S6 and pore helix) which appears to be rotated with respect to the VSLD. Indeed, the position of the pore domain of the apo human TRPV3 at the extracellular side aligns better with the pore domain of the open mTRPV3 channel than its apo counterpart. This might be indicative of inter-species differences, suggesting that the pore domain of the human TRPV3 channel does not undergo independent movement during gating (Supplementary Fig. 9). Alternatively, the structural divergence in the two apo structures might reflect the existence of multiple distinct apo, closed states in TRPV3.

Unexpectedly, the presence of 2-APB in the sample induces a departure from C4 symmetry within the channel. The twofold symmetric expansion of the gate in human TRPV3 was not observed in apo or sensitized TRPV3, and only manifested upon 2-APB addition, leading us to speculate that the two TRPV3$_{2\text{-APB}}$ structures we observe represent intermediate states associated with gating function. Intriguingly, normal mode analysis of the backbone (Cα) of the C4-symmetric channel, which was performed to estimate possible global protein conformations of TRPV3, resulted in trajectories that recapitulated the

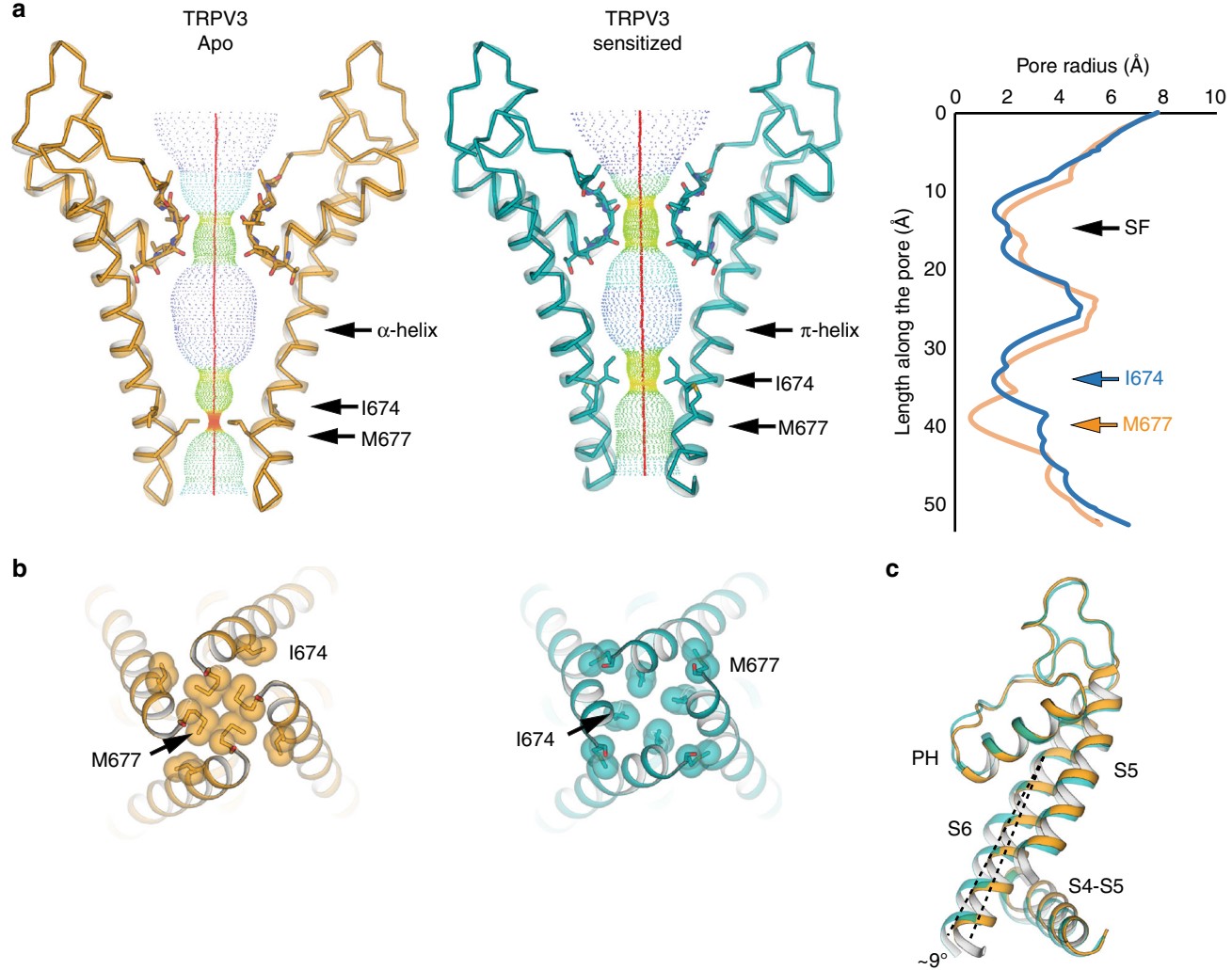

**Fig. 4** Comparison of apo and sensitized TRPV3. **a** Comparison of HOLE profiles of the apo (gold) and sensitized (cyan) TRPV3 shows that the TRPV3 channel has different gates in the two different functional states: M677 in the apo closed state, and I674 in the sensitized closed state. The difference stems from a register shift induced by the formation of a π-helix in the sensitized state. **b** Bottom-up view of the gate restrictions in the apo (gold, left) and sensitized (cyan, right) structures. **c** The π-helix induces a ~9° bend in the bottom part of S6 in the sensitized state

experimentally observed adoption of the C2-symmetric states (Supplementary Fig. 10). Our maps did not enable unambiguous identification of 2-APB molecules, making it difficult to ascertain if the observed C2 symmetry relates to incomplete binding of 2-APB to the channel. However, the recent structure of the closely related TRPV2 channel fully bound to RTx shows that twofold symmetric arrangements occur even when all ligand binding sites are occupied[36]. In addition, the observation of C2 symmetry in TRPV2 and TRPV3 suggests that the ability to enter C2-symmetric states during the gating cycle is a shared feature of thermoTRPV channels. Furthermore, recent cryo-EM studies of the structural transitions of the Na$^+$-activated K$^+$ channel Slo2.2 channel upon Na$^+$ titration did not reveal intermediate twofold symmetric arrangements[49], indicating that TRPV channels may utilize a fundamentally different gating mechanism from architecturally related K$^+$ channels.

## Methods

**Protein expression and purification of human TRPV3**. An expression screen of 32 homologs of TRPV3 using *Spodoptera frugiperda* 9 (Sf9) insect cells (ATCC) identified the human TRPV3 as the most suitable candidate for structural studies. To further stabilize hTRPV3, we searched a random mutant library of mouse TRPV3[3] for clones that exhibited either a large 2-APB response or a high baseline in a Ca$^{2+}$-dependent fluorescence assay. We aimed to find mutants with higher

expression levels due to increased stability. Fifty clones were sequenced, and we introduced 29 point mutations into their corresponding sites in the hTRPV3 (Supplementary Table 1). We expressed, purified, and tested the biochemical behavior of these point mutants using size-exclusion chromatography, and found that the T96A mutant showed an improvement in the monodispersity of the full-length human TRPV3 channel. The T96A mutation had a high rate of occurrence in tandem with other mutations, but did not appear on its own. Because in our patch clamp recording T96A does not affect the 2-APB response in hTRPV3, we reasoned that the enhanced calcium signals for the clones containing the T96A mutation was either due to the additional mutation present or the increase in the number of channels due to its enhanced stability by the T96A mutation.

A full-length human TRPV3 construct, containing mutation T96A was cloned into a pFastBac vector in frame with a FLAG affinity tag, and baculovirus was produced according to the manufacturers' protocol (Invitrogen, Bac-to-Bac). For protein expression, Sf9 insect cells (ATCC) were infected with baculovirus at a density of 1.3×10$^6$ cells ml$^{-1}$ and grown at 27 °C for 72 h in an orbital shaker. After 72 hours, cell pellets were collected, resuspended in buffer A (50 mM TRIS pH 8, 150 mM NaCl, 1 µg ml$^{-1}$ leupeptin, 1.5 µg ml$^{-1}$ pepstatin, 0.84 µg ml$^{-1}$ aprotinin, 0.3 mM phenylmethane sulfonyl fluoride, 14.3 mM β-mercaptoethanol, and DNAseI) and broken by sonication (3 × 30 pulses). 40 mM dodecyl β-maltoside (DDM, Anatrace) and 4 mM Cholesteryl Hemisuccinate Tris salt (CHS, Anatrace) were added to the lysate for extraction at 4 °C for 1 hour. Unsolubilized material was removed by centrifugation (8000 × g, 30 min), and anti-FLAG resin was added to the supernatant for 1 hour at 4 °C. For preparation of the apo sample, the resin was loaded onto a Biorad column at 4 °C and washed with ten column volumes buffer B (50 mM TRIS pH8, 150 mM NaCl, 1 mM DDM, 0.1 mM CHS, 10 mM dithiothreitol (DTT)) and the protein eluted in buffer C (50 mM TRIS pH 8, 150 mM NaCl, 1 mM DDM, 0.1 mM CHS, 0.1 mg ml$^{-1}$ 3:1:1 1-palmitoyl-2-

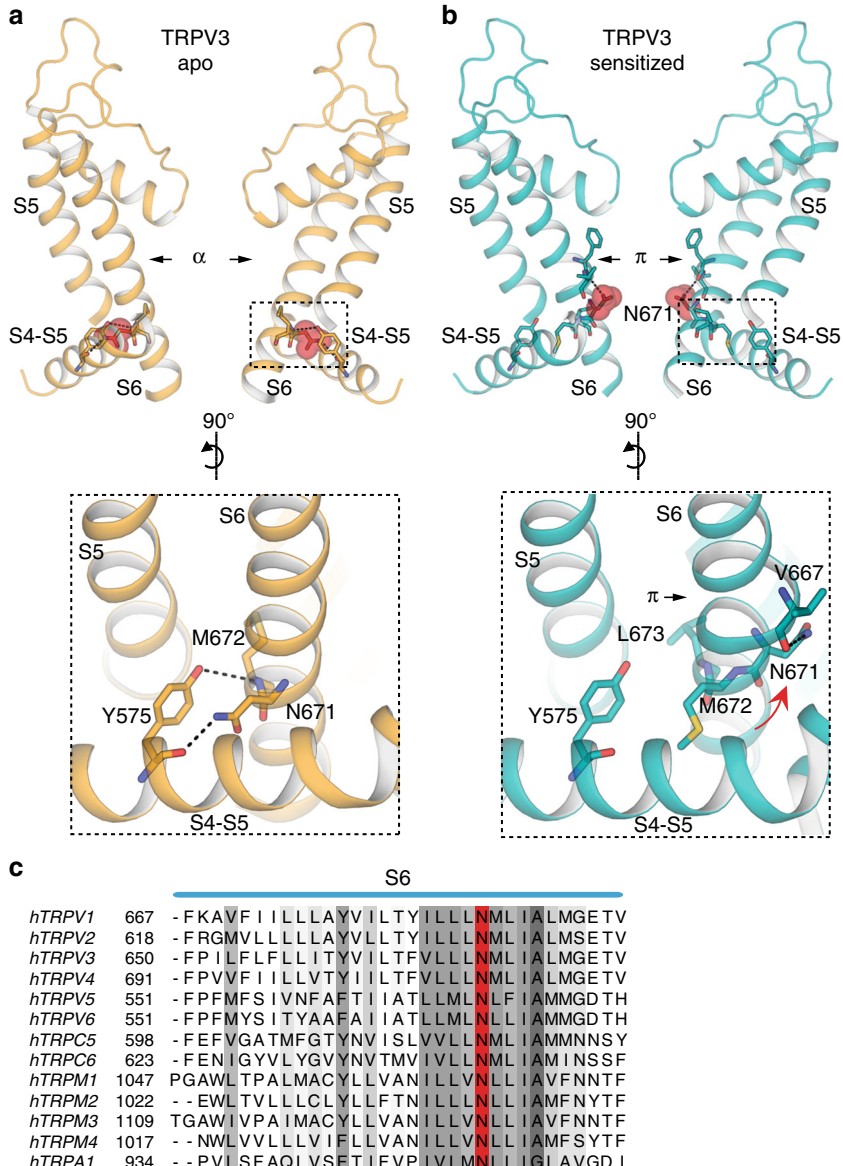

**Fig. 5** Formation of the π-helix in S6 changes the S4—S5 linker/pore domain interface. **a** The register of the α-helical S6 in the apo TRPV3 enables an interaction network between the Y575 in the S4—S5 linker and N671 and M672 in S6. Residue N671 is shown in sphere representation in top panel. **b** Formation of a π-helix in the sensitized state, changes the register of the helix, so that N671 turns away from the S4—S5 linker and into the pore, where it interacts with the backbone of V667 residue in the π-helix. Residue N671 is shown in sphere representation in top panel. **c** Sequence alignment of TRP channel S6 helices. The conserved N residue (N671 in TRPV3) is marked in red

oleoyl-*sn*-glycero-3-phosphocholine (POPC), 1-palmitoyl-2-oleoyl-*sn*-glycero-3-phosphoethanolamine (POPE), 1-palmitoyl-2-oleoyl-*sn*-glycero-3-phospho-(1′-*rac*-glycerol) (POPG), 10 mM DTT, 10 mg ml⁻¹ FLAG peptide). For preparation of the sensitized TRPV3, the anti-FLAG resin was loaded onto a Biorad column and washed with five column volumes Buffer B (50 mM TRIS pH 8, 150 mM NaCl, 1 mM DDM, 0.1 mM CHS, 10 mM DTT), followed by five column volumes Buffer B supplemented with 1 mM 2-APB. This was repeated ten times, before the protein was eluted in five column volumes buffer C (50 mM TRIS pH 8, 150 mM NaCl, 1 mM DDM, 0.1 mM CHS, 0.1 mg ml⁻¹ 3:1:1 POPC:POPE:POPG, 10 mM DTT, 10mg ml⁻¹ FLAG peptide). For preparation of the TRPV3$_{2\text{-APB}}$ sample, the same procedure was followed except buffer C was supplemented with 1 mM 2-APB. Following size exclusion chromatography, conducted at 4 °C, the protein peaks were collected, mixed with Poly (Maleic Anhydride-alt-1-Decene) substituted with 3-(Dimethylamino) Propylamine (PMAL-C8, Anatrace) (1:10 w/w ratio) and incubated overnight at 4 °C with gentle agitation. Detergent was removed with Bio-Beads SM-2 (15 mg ml⁻¹) for 1 hour at 4 °C. The reconstituted protein was further purified on a Superose 6 column at 4 °C in buffer D (50 mM Tris pH 8, 150 mM NaCl). In preparation of the TRPV3$_{2\text{-APB}}$ sample, Buffer D was supplemented with 1 mM 2-APB. Following size exclusion, the protein peak was collected and concentrated to 2–2.5 mg ml⁻¹.

**Cryo-EM sample preparation.** Cryo-EM grid preparation was performed similarly for each TRPV3 specimen. Briefly, 3 μl of purified TRPV3 (0.35–0.5 mg ml⁻¹) was dispensed on a freshly plasma cleaned (Gatan Solarus) UltrAuFoil R1.2/1.3 300-mesh grid (Electron Microscopy Services) and manually blotted[50] using a custom-built manual plunger housed in a 4 °C cold room (>85% relative humidity). Samples were blotted for 4–5 s with Whatman No. 1 filter paper immediately before plunge-freezing in liquid ethane cooled by liquid nitrogen.

**Cryo-EM data acquisition and image processing.** All microscope alignments were performed on a cross-grating calibration grid according to previously described methodologies[51] and all data were collected and preprocessed using the Leginon[52] and Appion[53] software packages, respectively.

Movies of apo TRPV3 were collected using a Thermo Fisher Titan Krios transmission electron microscope (TEM) operating at 300 keV with a Gatan K2 Summit direct electron detector (DED) operating in super-resolution mode (1.31 Å physical pixel size) at a nominal magnification of ×22,500. Movies were collected over a 12 s exposure with an exposure rate of ~9e- pixel⁻¹ s⁻¹, resulting in a total exposure of ~63e- Å⁻², using a defocus range of −0.5 to −2 μm. Motion correction was initially performed using Motioncorr[54] and the unweighted summed images

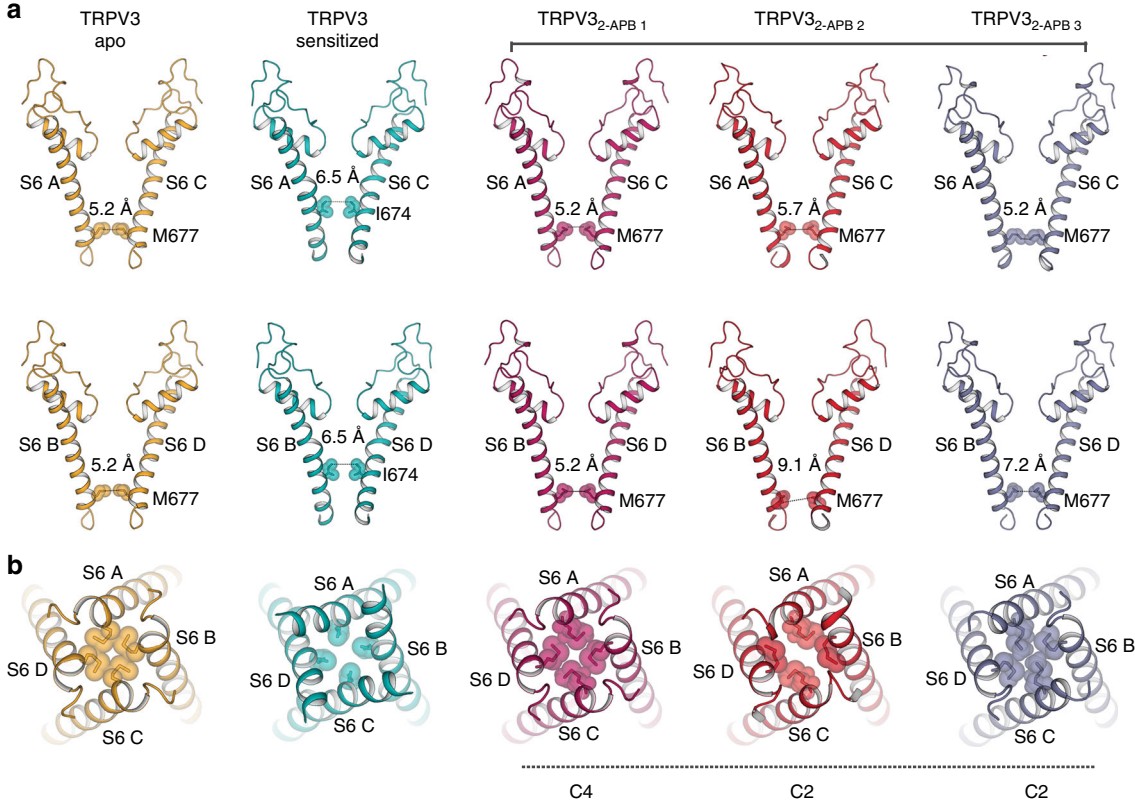

**Fig. 6** Overview of pores in apo, sensitized and TRPV3 in the presence of 2-APB. **a** Comparison of S6 helices of subunits A and C (top panel) and B and D (bottom panel) in apo (gold), sensitized (cyan), TRPV3$_{2\text{-APB }1–3}$ (magenta, red, and slate). Residues representing the helix bundle gate are shown as sticks and spheres. **b** Bottom-up view of the pores of apo (gold), sensitized (cyan), TRPV3$_{2\text{-APB }1–3}$ (magenta, red, and slate). Residues representing the helix bundle gate are shown as sticks and spheres

(Fourier binned 2 × 2) were used for CTF estimation using CTFFIND4[55] (whole micrograph CTF estimation) to remove those micrographs with a confidence of fit below 90% were eliminated from further processing. Difference of Gaussian (DoG) picker[56] was used to automatically pick particles from the first ~100 micrographs and then subjected to reference-free 2D classification using multivariant statistical analysis/multi-reference alignment (MSA/MRA)[57] (5.24 Å per pixel, 64-pixel box size). The best 2D classes that represented orthogonal views of TRPV3 were then used for template-based particle picking using FindEM[58]. A final stack of 361,244 particle picks were then subjected to reference-free 2D classification using RELION[59] version 1.4 (5.24 Å per pixel, 64-pixel box size) with the best 215,346 particles were subsequently 3D auto-refined using a 20-Å low-pass-filtered TRPV2 structure (EMDB-6455) as the initial model. Refined particles were then re-centered and re-extracted binned 2 × 2 (1.31 Å per pixel, 256-pixel box) and 3D auto-refined using a scaled version of the binned 8 × 8 map as the initial model. Particle movement and dose-weighting were then performed using the "particle polishing" feature as implemented within RELION. A smoothened plot of the per-frame B-factor and intercepts (Supplementary Fig. 1) were then used to calculate the frequency-dependent weighting. These "shiny" particles were then 3D auto-refined to yield a 3.95 Å resolution reconstruction (C1 symmetry). This reconstruction exhibited C4 symmetry. Subsequent no-alignment 3D classification using a mask generated against the full molecule low-pass filtered to 10 Å yielded a single good class that refined to 3.6 Å resolution (C4 symmetry). During the preparation of this manuscript the frame-alignment software MotionCor2[60] was released so the 282,136 particles contributing to the 3.6 Å resolution reconstruction were then re-centered and re-extracted from micrographs that had been aligned and dose-weighted using MotionCor2 without binning (0.655 Å per pixel) with per-particle CTF values as estimated using Gctf[61]. These particles were subjected to 3D classification where the best resolved class was subsequently 3D auto-refined to 3.5 Å resolution (C4 symmetry). These particles (120,424) were then subjected to a no-alignment 3D classification using a soft mask of the full molecule. The best resolving class (27,620 particles) was then 3D auto-refined (C4 symmetry) to a final resolution of 3.4 Å, as estimated by gold-standard FSC 0.143 criterion[62,63].

Movies of sensitized TRPV3 were collected on a Talos Arctica (Thermo Fisher) TEM operating at 200 keV. Movies were collected using a K2 Summit DED in counting mode at a nominal magnification of ×45,000 corresponding to a physical pixel size of 0.915 Å per pixel. A total of 2982 movies (47 frames/movie) were collected using a 12 s exposure with an exposure rate of ~4.7e- pixel$^{-1}$ s$^{-1}$, resulting in a total exposure of ~67 e- Å$^{-2}$ and a nominal defocus range from −0.8

to −1.4 μm. MotionCor2 was used to perform motion correction and dose-weighting. Unweighted summed images were used for CTF determination using CTFFIND4. FindEM was used for template-based particle picking using templates previously generated of apo TRPV3, yielding a stack of 559,206 picks that were binned 4 × 4 (3.66 Å per pixel, 112-pixel box size) and subjected to reference-free 2D classification using RELION 2.1[64]. A total of 339,693 particles corresponding to the 2D classes displaying the strongest secondary-structural elements were input to 3D auto-refinement in RELION without symmetry imposed. The apo TRPV3 volume was low-pass filtered to 30 Å and used as an initial model. The refined coordinates were used for re-centering and re-extraction of 2 × 2 binned particles (1.83 Å per pixel, 224-pixel box size) and subjected to 3D classification ($k = 6$) using a soft mask generated from the full molecule. A single class exhibiting C4 symmetry (44,553 particles) was 3D auto-refined with a soft mask and C4 symmetry applied to yield a ~3.7 Å reconstruction. These particles were re-centered and re-extracted unbinned (0.915 Å per pixel) and auto-refined to obtain a final resolution of ~3.24 Å (C4 symmetry).

Movies of TRPV3 in the presence of 2-APB were collected on a Talos Arctica equipped with a K2 Summit DED operating in counting mode. Images were collected at a nominal magnification of ×36,000 corresponding to a physical pixel size of 1.15 Å per pixel. A total of 1908 movies (60 frames per movie) were collected using a 15 s exposure with an exposure rate of ~5.3e- pixel$^{-1}$ s$^{-1}$, resulting in a total exposure of ~60e- Å$^{-2}$ and a nominal defocus range from −1.2 to −2 μm. All preprocessing (e.g. frame alignment, dose-weighting, CTF estimation, and particle picking) was performed as described above. A stack of 897,643 picks (binned 4 × 4, 4.8 Å per pixel, 72-pixel box size) were subjected to reference-free 2D classification in RELION. A subset of 589,656 particles corresponding to the best 2D classes were subsequently 3D auto-refined with C4 symmetry imposed, using the apo TRPV3 volume low-pass filtered to 30 Å as an initial model. The particles were then subjected to 3D classification ($k = 6$) without alignment, using a soft mask generated from the full molecule. A subset of 397,536 particles were selected and subsequently re-centered and re-extracted unbinned (1.15 Å per pixel, 288-pixel box size) and 3D auto-refined with C4 symmetry imposed to yield a ~3.6 Å reconstruction. An additional round of 3D classification ($k = 6$) without alignment and without symmetry imposed revealed one state exhibiting C4 symmetry and two distinct states exhibiting C2 symmetry, which subsequently refined to ~3.5 Å (TRPV3$_{2\text{-APB }1}$, C4 symmetry), ~4 Å (TRPV3$_{2\text{-APB }2}$, C2 symmetry), and ~4 Å (TRPV3$_{2\text{-APB }3}$, C2 symmetry), respectively. Local resolution estimates of the final reconstructions were calculated using BSOFT[65].

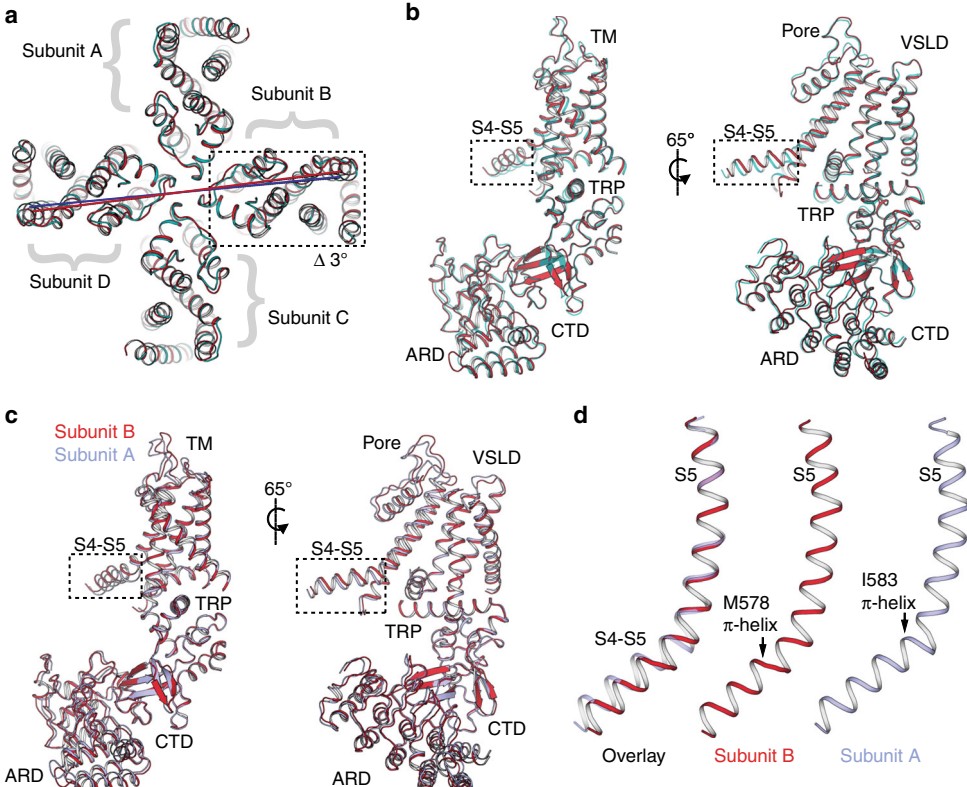

**Fig. 7** Reduced symmetry in the structures of TRPV3 obtained in the presence of 2-APB **a** Overlay of sensitized TRPV3 (cyan) and TRPV3$_{2-APB\,2}$ (red). Subunits B and D of TRPV3$_{2-APB\,2}$ undergo a ~3° rotation. **b** Superposition of subunit B of sensitized TRPV3 (cyan) and TRPV3$_{2-APB\,2}$ (red). The S4—S5 linker deviates from the alignment. **c** Overlay of subunits A (light blue) and B (red) of TRPV3$_{2-APB\,2}$. The S4—S5 linkers in the two subunits assume different conformations. **d** Overlay of S5 helices of subunits A (light blue) and B (red) of TRPV3$_{2-APB\,2}$. The two S4—S5 linkers have π-helices in different positions

**Model building and analysis**. The apo TRPV3 model was built in the cryo-EM electron density map in Coot[66], using the structure of TRPV2 (PDB 5AN8) as a template for the transmembrane domains and the structure of the human TRPV3 ankyrin repeat domains[41] (PDB 4N5Q) as a template for the ARD. The structure was real-space refined in Coot, with ideal geometry constraints where appropriate. The model was iteratively refined using the phenix.real_space_refine command line[67] by global minimization and rigid body, while maintaining high weight on ideal geometry and secondary structure restraints. Molprobity[68] (http://molprobity. biochem.duke.edu/) was used to identify problematic regions, which were subsequently rebuilt in Coot. The refined apo TRPV3 model was used as a template for building the remaining four models. Radius along the permeation pathway was calculated using HOLE[69]. All analysis and structure illustrations were performed using Pymol (The PyMOL Molecular Graphics System, Version 2.0) and UCSF Chimera[70].

**Normal mode analysis**. Normal mode analysis was performed on the apo TRPV3 structure using the backbone (Cα) atoms using the program ProDy[71] as recommended by the developers.

**Electrophysiology**. HEK293T cells (62312975—ATCC) were grown in Dulbecco's Modified Eagle Medium supplemented with 10% fetal bovine serum (Gibco), 1% penicillin/streptomycin (Gibco) and were sustained at 37 °C in 5% CO$_2$. Cells between passage 10–30 grown in 40-mm wells were transiently transfected at ~50% confluency with plasmids encoding for either WT or T96A TRPV3 and green fluorescent protein using FuGene6 (Promega). Twenty-four hours post transfection, cells were reseeded onto 12-mm round glass coverslips (Fisher) and used 12–24 hours after for electrophysiological measurements.

Voltage-clamp recording were performed at room temperature (22 °C) in the whole-cell patch configuration with electrodes pulled from borosilated glass capillaries (Sutter Instruments) with a final resistance of 2.5–5 MΩ. Electrodes were filled with an intracellular solution containing (in mM) 150 CsCl, 1 MgCl$_2$, 10 4-(2-hydroxyethyl)-1-piperazineethanesulfonic acid (HEPES), 5 ethylene glycol-bis(β-aminoethyl ether)-N,N,N′,N′-tetraacetic acid, and adjusted to pH 7.2 (CsOH). Glass coverslips with adherent transfected cells were placed into an open bath chamber (RC-26G, Warner Instruments) and an extracellular solution containing (in mM) 140 NaCl, 5 KCl, 1 MgCl$_2$, 10 HEPES at pH 7.4 (NaOH) with and without (wash) 30 μM 2-APB (Sigma) (prepared daily from dimethyl sulfoxide (DMSO)

stocks (1 M) stored at −80 °C; final DMSO 0.03%) was focally applied with a pressurized perfusion system (BPS-8, ALA Scientific Instruments).

2-APB sensitization experiments were performed with a 30-s continuously repeating protocol (holding potential of +60 mV) in which the cell was first perfused with extracellular wash for 1 s, followed by a 15-s application of 2-APB extracellular solution, then preceded by 14 s of wash solution. The repeating recording protocol was then stopped when the 2-APB-elicited responses no longer exhibited potentiation of the peak current amplitude. Current responses were low-pass filtered at 2 kHz (Axopatch 200B), digitally sampled at 5–10 kHz (Digidata 1440A), converted to digital files in Clampex10.7 (Molecular Devices) and stored on an external hard drive for offline analyses (Clampfit10.7, Molecular Devices; OriginPro 2016, OrginLab Corp.)

**Data analysis**. Sensitization parameters were evaluated by the stimulation-dependent increase in the peak current amplitude measured at the end of each 2-APB exposure. Current amplitudes ($I$) from each stimulation where normalized to the maximum current ($I_{max}$) amplitude, and the fractional current ($I/I_{max}$) of each stimulation was plotted by stimulation number for each individual recording[13]. The stimulation time course of sensitization was fitted with a standard two-state Boltzmann equation:

$$\frac{I}{I_{max}} = \frac{1}{1 + e^{[(S-S_{50})/k]}} \qquad (1)$$

with $S$ the stimulation number, $S_{50}$ the stimulation number corresponding to $I = I_{max}/2$, and the slope factor $k$, a fitted constant expressed in stimulation number representing the rate of change of sensitization. The fitted parameter values were obtained by fitting individual $I/I_{max}$ vs. stimulation plots from biologically independent experiments and individual values were used to calculate the mean $S_{50}$ and $k$ for WT and T96A. The extent of sensitization was characterized by the relative increase in current obtained during the first ($I_0$) and maximum current ($I_{max}$) stimulation ($I_{max}/I_0$) and calculated as the mean from each biologically independent experiment.

## Data availability

Data supporting the findings of this manuscript are available from the corresponding authors upon reasonable request. The sequence of human TRPV3 can be found in the National Center for Biotechnology Information (accession code

NM_145068.3). Coordinates have been deposited in the Protein Data Bank under accession codes 6MHO, 6MHS, 6MHV, 6MHW, and 6MHX. Cryo-EM density maps have been deposited in the Electron Microscopy Data Bank under accession codes EMD-9115, EMD-9117, EMD-9119, EMD-9120, and EMD-9121. The source data underlying Fig. 1e–g are provided as a Source Data file.

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

## Acknowledgements

All cryo-EM data were collected at The Scripps Research Institute (TSRI) electron microscopy facility. We thank A. Patapoutian at TSRI for his generosity in providing access to his mutant TRPV3 library. We thank J. Grandl at Duke for the guidance with mutant selection and analyses. We thank J.C. Ducom at TSRI High Performance Computing facility for computational support and B. Anderson for microscope support. We thank S. Thomas for initial TRPV3 biochemistry and S.E. Jordt and S. Jabba at Duke for their help with preliminary calcium imaging. This work was supported by the National Institutes of Health (R35NS097241 to S.-Y.L., DP2EB020402 and R21AR072910 to G.C.L.). G.C.L. is supported as a Searle Scholar and a Pew Scholar in the Biomedical Sciences, supported by the Pew Charitable Trusts. M.A.H. is supported by a Helen Hay Whitney Foundation postdoctoral fellowship. M.W. is supported by the National Science Foundation Graduate Research Fellowship program. Computational analyses of EM data were performed using shared instrumentation at TSRI funded by NIH S10OD021634.

## Author contributions

L.Z. conducted biochemical experiments of TRPV3 and model building and refinement, W.F.B. carried out electrophysiological recordings, M.H. conducted sequencing of mouse TRPV3 clones and initial TRPV3 biochemistry, all under the guidance of S.-Y.L. S.-Y.L. conceived the idea to isolate stabilizing mutations from the TRPV3 mutant library. A.S.S. performed the normal mode analysis under the guidance of M.A.H. M.A.H. and M.W. conducted all cryo-EM experiments under the guidance of G.C.L. L.Z. S.-Y.L., G.C.L., M.A.H. and M.W. wrote the paper.

## Additional information

**Competing interests:** The authors declare no competing interests

