## [Peer Review File · Nature Communications]

This manuscript has been previously reviewed at another journal that is not operating a transparent peer review scheme. This document only contains reviewer comments and rebuttal letters for versions considered at Nature Communications. Mentions of prior referee reports have been redacted.

We thank the reviewers for their enthusiasm about our studies and constructive criticisms. We have revised our manuscript per reviewers' comments. For clarity, our responses are italicized.

Reviewer #1 (Remarks to the Author):

TRPV3, a thermo-sensitive TRP channel, serves as a modulator for biological events in epidermal and hair follicle keratinocytes. The authors found a mutation T69A that does not affect the function of TRPV3 but significantly stabilizes the protein for structural investigations. This tour de force allows the authors to capture the TRPV3 in the distinct states at the atomic level. In general, this manuscript presents an interesting and exciting story that illuminates the molecular mechanism of TRPV3 opening and regulation. This referee also appreciates that the authors posted this manuscript to bioRxiv (doi: <https://doi.org/10.1101/395616>) before it is accepted for publication. This finding seems to warrant publication in Nature Communications in this referee's opinion, since the work will serve to present an intact story and illuminate the regulatory mechanism of this important channel. This work should be accepted, after the authors address the following concerns.

Major Concerns:

1, the authors points out that N671, the only polar residue in S6, plays a key role in regulating channel activity. It is necessary to introduce a mutation on N671 to perform functional characterization of this mutant. This will convince the readers of the physiological importance of this residue.

R) We acknowledge that a functional characterization of the N671 residue would increase the importance of this residue. However, the purpose of pointing out the change in position of N671 was merely to point out that the α -to- π transition in the S6 helix changes the electrostatic properties of the pore. Other groups have identified a similar role for the corresponding residue in TRPV1 (Kasimova et al, J Phys Chem Lett, 2018; Kasimova et al, Biorxiv, DOI <https://doi.org/10.1101/310151>, 2018), and we have now included references to these papers in the manuscript.

2, since this manuscript reported an opening mechanism of TRPV channels, the authors could consider comparing the opening of TRPV3 with other types of TRP channels (McGoldrick et al, Nature for TRPV6, Schmiege et al, Nature for TRPML1; Su et al Nature Communications for PKD2L1).

R) Unfortunately, we have not managed to capture a fully open state of the human TRPV3 channel. Instead, we have presented a series of putative intermediate states. Therefore, we have been cautious in our interpretation and have not described an opening mechanism but have instead focused on describing the structural elements which undergo conformational change in our various experimental conditions. Since we do not provide a full mechanistic trajectory of gating, we have not felt that it is appropriate to compare our structures to previously determined gating mechanisms in other TRP channels. However, we believe that the data we present here has offered us a unique opportunity to inspect the various intermediate conformations that the TRPV3 channel can adopt,

and the presence of C2 symmetry has enabled us to dissect how distinct domains (i.e. ARD and TMD) might be coupled.

3, update the discussion to include a brief comparison with recently published details of TRPV3 (Singh et al NSMB). There are similarities and differences.

R) We have now included a section in the discussion as well as a supplementary figure (Supplementary Fig. 9) where we compare the mouse TRPV3 to human TRPV3 structures.

Minor Points:

4, Line 72, the small-molecule inhibitor should be indicated.

R) We apologize for this omission, and have now included the names of the small-molecule inhibitors in the manuscript.

5, Line 344, does the buffer C contain Flag peptide?

R) We thank the reviewer for pointing out this omission. Buffer C does indeed include Flag peptide. We have added this to the methods.

6, Line 137-138 they refer to suppl Figure 2 and it may be Figure 4. Honestly the authors should probably point this out earlier in the results section where they are defining the channel activity. This referee feels there is room for the three bar graphs from Supplemental Fig 4 to be placed into Fig. 1.

R) We have updated Fig. 1 to include the three bar graphs from Supplementary Fig. 4 and have corrected the figure reference in line 137-138.

7, mention temperature please in methods. Perhaps show a temperature of camphor control for WT vs T96A.

R) We have now updated the methods to include temperature conditions for protein preparation, grid freezing and electrophysiology experiments. We agree that temperature and camphor controls would be a nice addition. However, since our study focuses on ligand-dependent activation, we feel that temperature experiments are beyond its scope. In the study, we have chosen 2-APB as a ligand as it is the most potent and well-studied agonist of TRPV3. Camphor control experiment would be nice, but it is likely that camphor binds to a site (or sites) in TRPV3 different from 2-APB. In addition, the recently published NSMB paper on TRPV3 (Singh et al, NSMB, 2018) does not report temperature or camphor responses for the mutant channels presented in the paper.

Reviewer #2 (Remarks to the Author):

Zubcevic et al. present cryo-EM structures of human TRPV3, both in the apo state and in several 2-

APB stimulated states. The data appear of high quality, and provide novel insights into TRPV3 channel functioning.

Specific points:

1) To obtain structures of the sensitized TRPV3, the authors apply a protocol with repeated incubations with 2-APB followed by washing steps to TRPV3 attached to anti-FLAG resin. Whereas the authors show that such a protocol can sensitize the channel when present in a cell membrane, this does not necessarily mean that similar sensitization occurs in a cell-free environment in a Biorad column. Is it known whether TRPV3 sensitization actually occurs in a cell-free environment (e.g. in lipid bilayers)? In the absence of such evidence, it seems premature to denote this structure as “sensitized” TRPV3, and the authors should at least discuss this in more detail.

R) We accept the reviewer’s criticism. In our revised manuscript we refer to this structure as “putative sensitized”. Nevertheless, application of the 2-APB sensitization protocol resulted in a structure that is distinct from both the human apo TRPV3 and the open mouse TRPV3 structures, in that it possesses a π -helical turn in S6 but its gate is closed. Therefore, our putative sensitized structure is clearly an intermediate state that is on the path toward the open state, consistent with the sensitized states. However, we cannot say that the putative sensitized structure presented here is the fully sensitized state of TRPV3. Studies suggest (Liu et al, JGP, 2011) that TRPV3 might have multiple such sensitized states, with decreasing energy requirements for opening. In addition, structural elements other than S6 might be undergoing conformational changes during sensitization (Liu et al, PNAS, 2017). We have now included a section in the discussion where this is addressed in more detail.

2) The authors present several structures of TRPV3 in the presence of 2-APB, which no longer show fourfold symmetry. Have the authors considered the possibility that these could arise from channels in which the 2-APB ligand binding sites are only partially occupied? Channels with 1-3 2-APB molecules bound would automatically lose fourfold symmetry. Or do the authors have evidence that all four ligand binding sites are occupied in the structures?

R) This is an interesting point. Unfortunately, we could not unambiguously identify 2-APB in our cryo-EM maps, so we cannot say with certainty whether the 2-APB binding sites are fully occupied, or how many 2-APB molecules might be bound per tetramer. Our experiments were performed in the presence of high 2-APB (1mM), which would make partial occupancy less likely but not impossible. Interestingly, our previous work on TRPV2 (Zubcevic et al, NSMB, 2018) showed that C2 symmetry can be achieved even when the binding sites are fully occupied. In this case, the binding mode of the ligand differs in diagonally opposing subunits, resulting in a two-fold symmetric channel. We have now included a section in the discussion that addresses this question.

3) Singh et al. recently published TRPV3 structures in the apo and agonist-bound states (NSMB, 2018). Please discuss/compare.

R) We have now included a section in the discussion as well as a supplementary figure (Supplementary Fig. 9) where we compare the mouse TRPV3 to human TRPV3 structures.

4) In the introduction:

Line 54-56: The statement that TRPV1-TRPV4 are involved in thermosensing is not supported by the literature. Whereas the role of TRPV1 as one of the sensors of noxious heat is well established, elimination of TRPV2, TRPV3 or TRPV4 in mice does not have any effect on thermosensation. Please reword, and/or cite more recent work or reviews on this topic

R) The reviewer brings up a valid point, and it seems that we have unintentionally miscommunicated the statement on thermosensation. Our comment related to the ability of the channels to open in response to heat, which is well documented for TRPV1-TRPV4, and not their physiological role in thermosensation. We have now reworded this sentence.

Line 59: reference 12 seems out of place here, and references 1 and 11 are a bit outdated and do not really provide an up to date view on the role of TRPV channels.

R) We accept this and have updated the references to reflect the pharmacological and biophysical diversity in the thermoTRPV channels.

Reviewer #3 (Remarks to the Author):

TRPV3 plays important roles in skin, hair and itch physiology. Gain of function, eg due to mutations in the S4-S5 linker, results skin disease called Olmsted. Loss of function, eg via inhibitors applications, may be associated with analgesic effects on inflammation and pain. TRPV3 can be activated by repeated application of heat or agonists such as 2-APB, and is regulated by lipids such as PIP2. This study reports high resolution Cryo-EM structures of human full-length TRPV3 bearing mutation T96A.

This manuscript from Dr Lee's group presents high quality structural data that show apo and sensitized (or activated) states of TRPV3 channels, as well as three 2-APB bound 'intermediate' states. The finding of decreased symmetry to C2 is novel and intriguing. The authors found that the T96A mutant is optimal as compared with WT protein for structural purposes and that it has similar functional characteristics as WT channel.

Major comments.

1. I agree that Fig. 1c and d and Suppl Fig. 4 show that the T96A mutation does not affect the basic channel property. However, I do not agree with the use of 'hysteresis' in line 98. Hysteresis is characterized by the presence of a so-called hysteresis loop/curve, ie, for a given x value (eg temperature, agonist concentration, or repeated # of a given agonist concentration) there are two different y values, obtained from measurements carried out with increasing and decreasing the x

value, respectively. Data in Fig 1c and d rather show 'use-dependence', a term previously used (PMID: 28154143). To show that mutation T96A does not affect the hysteresis, the authors should use both increasing and decreasing temperature to obtain two temperature-dependence curves that form a hysteresis type of loop (ie non-superposing curves), as reported in 2002 for TRPV3 (PMID: 12077604).

R) We agree with the reviewer, and have therefore exchanged the term “hysteresis” for “use-dependence”. Because this study focuses on ligand-dependent activation of TRPV3, we feel that temperature experiments, while a nice addition, are beyond its scope and would not change the conclusion of our paper.

2. Line 341, the sensitization protocol used for preparing sensitized proteins for structural purposes. This protocol is very different from the one used in electrophysiology recordings. One key difference between the two is with or without voltage clamp, which would lead to different channel activities and possibly also agonist binding affinities. I'm not sure whether this protocol would effectively sensitize TRPV3 channels. To prove, the authors have to mimic this protocol in electrophysiology recordings to determine whether it indeed results in similar channel sensitization. To do this, a cell under the whole-cell mode should first be recorded for apo currents, and then de-clamped to undergo repeated agonist loads/washes (as described around line 341). The cell will then be recorded for TRPV3 currents to determine whether there is a similar sensitization/activation as in Fig. 1c and d.

R) We accept the reviewer’s criticism. In our revised manuscript we refer to this structure as “putative sensitized”. Nevertheless, application of the 2-APB sensitization protocol resulted in a structure that is distinct from both the human apo TRPV3 and the open mouse TRPV3 structures, in that it possesses a π -helical turn in S6 but its gate is closed. Therefore, our putative sensitized structure is clearly an intermediate state that is on the path toward the open state, consistent with the sensitized states. However, we cannot say that the putative sensitized structure presented here is the fully sensitized state of TRPV3. Studies suggest (Liu et al, JGP, 2011) that TRPV3 might have multiple such sensitized states, with decreasing energy requirements for opening. In addition, structural elements other than S6 might be undergoing conformational changes during sensitization (Liu et al, PNAS, 2017). We have now included a section in the discussion where this is addressed in more detail.

3. Line 272, “We propose that the transition from α - to π -helix is an essential component of TRPV3 hysteresis.” Need justifications for this statement. This is because the opposite π - to α -helix transition was reported for other TRPs such as V1, P2 and P3, but there is no evidence of hysteresis associated with these channels. It would be of help if the authors justify the statement through comparing with this opposite transition associated with (equivalent) channel activation. This point links to another statement (line 165), which may have to be changed: TRPV1 is also a thermoTRP but it has an apposite π - to α -helix transition.

R) We agree with the reviewer that more discussion concerning the α -to- π transition and sensitization should be included. Unfortunately, we are not familiar with the studies that show that

TRPV1 undergoes a π -to- α transition during opening. Work from Julius and Chen labs has shown that TRPV1 has a π -helix in S6 in both apo/closed and toxin bound/open states (Liao et al, Nature, 2013; Cao et al, Nature, 2013; Gao et al, Nature, 2016), suggesting that no secondary structure transitions occur in S6 of the TRPV1 channel during gating. This correlates well with the fact that, in contrast to TRPV2 and TRPV3, TRPV1 does not exhibit heat or capsaicin induced use-dependence (Liu et al, Biophys J., 2016). We have now included a section in the discussion to address this.

Indeed, a π -to- α transition has been associated with opening of the TRPP3 (PKD2L1) channel (Su et al, Nat Comms, 2018). In this study, the authors compare the closed structure of TRPP2 (PKD2), which contains a π -helix, with the open structure of TRPP3, which has an α -helical S6 and deduce that opening results from π -to- α secondary structure transitions in S6. Such π -to- α transitions have not been observed in the related TRPML channels (Zhou et al, NSMB, 2017; Schmiege et al, Nature, 2017), where the channels maintain a π -helix in both closed and open states. However, TRPP and TRPML channels differ substantially from the TRPV subfamily in both structure and function, and we feel that including a direct comparison here may not give much structural insights into TRPV channel gating.

Minor comments.

1. As mouse TRPV3 structure was just published in Nat Struct Mol Biol, I invite the authors to cite the paper and provide a comparison discussion.

R) We have now included a section in the discussion as well as a supplementary figure (Supplementary Fig. 9) where we compare the mouse TRPV3 to human TRPV3 structures.

2. Line 147, “the sensitized structure is bent at a $\sim 9^\circ$ angle towards the inner pore when compared to the S6 in the apo structure”. To me, this would mean that the pore will become smaller but in the reality it’s opposite (Fig. 4a right panel). A clearer sentence would help.

R) We agree with the reviewer that the sentence is confusing. We have now rewritten it, and it reads: “In addition, the C-terminal half of the S6 helix in this structure is bent at a $\sim 9^\circ$ angle from the inner pore when compared to the S6 in the apo structure (Fig. 4c).”

3. Does sensitized TRPV3 has a 2-APB molecule(s) bound? If yes, how many 2-APB molecules would be present? For 2-APB bound states 1-3, I would expect to see a speculation on whether they may correspond to distinct number of 2-APB molecules bound to one tetrameric channel. If this is not the case, then what variable could have allowed the formation of these states?

R) The reviewer raises an interesting point. However, we could not unambiguously identify 2-APB in our cryo-EM maps, so we cannot say with certainty whether the 2-APB binding sites are fully occupied, or how many 2-APB molecules might be bound per tetramer. Our experiments were performed in the presence of high 2-APB (1mM), which would make partial occupancy less likely but not impossible. Interestingly, our previous work on TRPV2 (Zubcevic et al, NSMB, 2018) showed that C2 symmetry can be achieved even when the binding sites are fully occupied. In this case, the binding

mode of the ligand differs in diagonally opposing subunits, resulting in a two-fold symmetric channel. We have now included a section in the discussion that addresses this question.

4. Spelling error in Abstract, line 35, simulation must be stimulation.

R) We thank the reviewer for pointing out this typo. It has now been corrected.

Reviewers' Comments:

Reviewer #1:

Remarks to the Author:

The concerns of this referee have been addressed by authors; therefore, this referee supports the revised manuscript for publication.

Reviewer #2:

Remarks to the Author:

The authors have adequately reviewed the manuscript.

Reviewer #3:

Remarks to the Author:

The authors have satisfactorily addressed most of my comments. For my comments regarding π and α transition in S6, I would like to point out 1) that TRPV6 has transition from α - to π -helix transition in S6 from a closed state to an open state (Saotome et al, Nature, 2016) but does not seem to link to use-dependence, and 2) that TRPP2 S6 helix undergoes π - to α - transition from apo (WT TRPP2) to an open state (TRPP2 F604P mutant) (Zheng et al, Nat Commun, 2018). Thus, π to α or α to π transition is shared by several TRP channels and the list may continue to increase. The statement (line 286) that there is an apparent correlation between S6 α to π transition and use-dependence can't hold.

We thank all reviewers for their constructive input in this manuscript. Also, we thank reviewer #3 for engaging in a discussion about secondary structure transitions in S6 of TRP channels during gating. Our response can be found below.

Reviewer #3 (Remarks to the Author):

The authors have satisfactorily addressed most of my comments. For my comments regarding π and α transition in S6, I would like to point out 1) that TRPV6 has transition from α - to π -helix transition in S6 from a closed state to an open state (Saotome et al, Nature, 2016) but does not seem to link to use-dependence, and 2) that TRPP2 S6 helix undergoes π - to α - transition from apo (WT TRPP2) to an open state (TRPP2 F604P mutant) (Zheng et al, Nat Commun, 2018). Thus, π to α or α to π transition is shared by several TRP channels and the list may continue to increase. The statement (line 286) that there is an apparent correlation between S6 α to π transition and use-dependence can't hold.

R. We agree with the reviewer that secondary structure transitions in S6 are fundamental to the gating of TRP channels. Here we are sharing our observation that the α -to- π transition appears to correlate with thermoTRPV channels that exhibit use-dependence. We view use-dependence as a reflection of the channel's trajectory through the energetic landscape, where the channel travels from a low-energy closed state to a high-energy open state through a series of intermediate energy points. The distinct architecture of individual channels determines their different conformational landscapes. TRPV6 is a constitutively open channel where the π -helical S6 and open gate represents the starting point. Because its energy landscape is distinct compared to thermoTRPVs, we believe that the "gating" of this channel would be different from TRPV3. We can only speculate that the underlying cause for this difference might be in their unique designs, specific interactions with lipids or it might be related to the fact that TRPV6 is not activated by temperature or ligand. In the case of TRPP2, it appears that the introduced proline mutation (F604P) in S5 leads to a secondary structural change in S5, which in turn causes the π -to- α transition in S6, as the mutation site in S5 is adjacent to the π -to- α transition site in S6. Therefore, the π -to- α transition in S6 is compensatory to the mutation in S5, not due to ligand-dependent activation. It is noteworthy that the π -to- α (or α -to- π) transitions in TRPP2 and human TRPV6 channels are obtained due to the mutations, and are not ligand induced.

Secondary structure transitions are a fascinating phenomenon, but we are only in the very early phases of understanding their role in the physiology of TRP channels. To acknowledge this, we have now added a section in the discussion that further study is needed to probe their role in use-dependence.